# Immunological and Biochemical Interplay between Cytokines, Oxidative Stress and Schistosomiasis

**DOI:** 10.3390/ijms22137216

**Published:** 2021-07-05

**Authors:** Priscilla Masamba, Abidemi Paul Kappo

**Affiliations:** Molecular Biophysics and Structural Biology (MBSB) Group, Department of Biochemistry, Kingsway Campus, University of Johannesburg, Auckland Park, Johannesburg 2006, South Africa; presh4u@rocketmail.com

**Keywords:** cytokines, dendritic cells, eosinophils, granuloma, macrophages, schistosomiasis

## Abstract

The host–parasite schistosome relationship relies heavily on the interplay between the strategies imposed by the schistosome worm and the defense mechanisms the host uses to counter the line of attack of the parasite. The ultimate goal of the schistosome parasite entails five important steps: evade elimination tactics, survive within the human host, develop into adult forms, propagate in large numbers, and transmit from one host to the next. The aim of the parasitized host on the other hand is either to cure or limit infection. Therefore, it is a battle between two conflicting aspirations. From the host’s standpoint, infection accompanies a plethora of immunological consequences; some are set in place to defend the host, while most end up promoting chronic disease, which ultimately crosses paths with oxidative stress and cancer. Understanding these networks provides attractive opportunities for anti-schistosome therapeutic development. Hence, this review discusses the mechanisms by which schistosomes modulate the human immune response with ultimate links to oxidative stress and genetic instability.

## 1. Introduction

The complexities of the inter-relationship between the mammalian host and the schistosome parasite have a significant bearing on the human immune response and the redox network. It is a well-known fact that the aggressive entry of helminth parasites into the human host system is primarily intended for colonization and long-term survival through immunomodulation of the immune system. Over eons of evolutionary time schistosomes have successfully developed the ability to stealthily plague the host and evade detection, not necessarily by multiplying in number or fluctuating antigen homeostasis, but by causing tolerance of the immune system to worm antigens instead, resulting in the host’s inability to eliminate these parasites [1]. Many of the pathologies and a vast bulk of the morbidity resulting from schistosome infection are mainly attributed to immune response-based inflammation against eggs caught in host tissues. Hypothetically speaking, schistosomiasis control and elimination should be effortless because of the multiple but treatable lifecycle stages of the parasite. However, association of the parasite with the human host has significantly augmented its persistence over time [2]. Regardless of the countless antigen molecules tested for vaccine development, none has been successful to date. Wilson and Coulson rightly declare that “… schistosomes are not stupid. They have had tens of millions of years to evolve mechanisms that help them survive immune attack from the mammalian host, even an attack orchestrated by our vaccine strategies. Schistosomes are … a formidable adversary that won’t be easily beaten” [3]. To a great extent, most of the human host’s schistosome-acquired immunity is attributed to antibody-dependent mechanisms. The longevity of schistosomes within the human host, like so many other helminths, is accompanied by a myriad of immune implications, which the parasite has been able to manipulate successfully and modulate for its own survival. While down-regulation of the immune response not only awards the parasite the opportunity to develop and thrive in the host, a dual effect is also presented as evolution of the host-parasite interaction limits severe pathology by modulating vigorous immunopathology [4]. It has been thought for a long time that immune modulation is almost always induced by eggs produced by adult worms, but growing evidence now supports the idea that immune regulation actually begins from the point of infection throughout all the stages to which the host is exposed. Biomolecules secreted by skin-penetrating cercariae, migrating schistosomulae, larval and adult worms, together with their eggs, are able to modulate both innate and adaptive immune responses, bestowing on the schistosome parasite evasion tactics from the host’s defense mechanisms. These actions include the down-regulation or up-regulation of cytokines that either promote or inhibit inflammation respective, as well as switches between the Type 1 T helper (Th1) and Type 2 T helper (Th2) immune response [5]. Once infected with the parasite, the initial response within infected tissues and the plasma is characterized by type-1 inflammation, which is driven by IL-1, IL-12, tumor necrosis factor-α and interferon-γ. This response is subsequently stifled during chronic schistosomiasis by the CD4^+^ T helper 2 (Th2) response, which is triggered by antigens secreted by the eggs and is driven by IL-4, IL-5, IL-10, and IL-13 (Figure 1) [6].

While the host battles with issues pertaining to immunity, the presence of the granuloma, which is typically associated with schistosomiasis immunity, is in addition highly connected to oxidative damage through the generation of free radicals. The production of reactive oxygen species (ROS) is an inevitable process of cellular aerobic metabolism but may be aggravated by ionizing radiation, UV light, exercise, pollutants, as well as various physiological disorders such as HIV/AIDS, cancer, diabetes, and helminth infections [7,8]. Free radicals are extremely reactive molecules formed by unmatched electrons in the valence electron shell that have the ability to react with or generate other free radicals by subtracting an electron from neighboring molecules, causing their instability. This creates a rippling effect where incomplete molecules seek the “missing” electron in an attempt to stabilize themselves, but simultaneously cause irreversible damage to and the modification of various cellular components such as membranes and receptors [9]. While in the human, both the host and schistosome parasite experience immense oxidative stress from the release of free radicals either generated by the human host through the immune response or from the parasite itself via respiration, resulting in the production of ferrous iron (Fe^2+^) and toxic heme through the consumption and breakdown of hemoglobin (Hb) [10]. Surprisingly, the schistosome parasite is poorly equipped to handle oxidative stress, but despite these unfavorable circumstances, the worm has devised persistent means to survive and occupy the human host for even up to three decades without experiencing any severe effects [11]. Broadening and expanding knowledge in these areas creates several opportunities for anti-schistosome therapeutic development. Hence this review discusses the various strategies the schistosome worm uses to trigger the immune system to not only establish its production, growth and thriving ability, but in the process, protect the host from severe disease and consequently prolong its survival.

## 2. Pathogenesis Associated with Schistosomiasis

In its early stages, schistosomiasis is considered an asymptomatic disease that is due to a toxemic and allergic reaction or systemic hypersensitivity to migrating and maturing larvae, while its acuteness is rather a result of worm burden and immune response to parasite antigens [12,13]. The complex migratory lifecycle of the schistosome has been described in great detail in literature and is initiated by the cutaneous penetration of free-swimming multicellular infective cercariae into the human mammalian host, aided by a number of environmental signals such as light-dark contrast, motion, chemical, and thermal gradients [12]. These concurrently lose their bifurcated tails to develop into schistosomulae [14]. Contact with cercariae-infested waters for a minimum of 1–5 min is all that is required for infection, which typically happens among especially young adolescents in endemic regions [13,15]. Cercarial dermatitis, commonly referred to as Katayama fever, is a form of rash that may appear 24 h after exposure and disappears again within a few hours. This is referred to as the acute phase and is mostly associated with the Th1 response and high levels of tumor necrosis factor (TNF-α) and interferon gamma (IFN-γ) (Figure 1) [16]. However, continuous exposure leads to the production of other pro-inflammatory cytokines that include IL-1β, IL-6, IL-10, IL-12, and M1PI1α [4]. The schistosome infective agent dictates varying symptoms that include dry cough and angio-edema, headache, fever, urticaria, and maculopapular pruritic skin eruption (described above as cercarial dermatitis), typically observed among first-time non-immune travelers to schistosome-endemic areas [17]. 

These symptoms may, however, digress into a number of more serious and complicated clinical sequelae. It is the intention, from the schistosome’s perspective, that its eggs be expelled through urine and feces based on the various schistosome species location. Target sites include the rectum and distal colon for *S. japonicum*, *S. mansoni*, *S. intercalatum*, and *S. mekongi* parasites, while the bladder is the location of choice for *S. haematobium* worms [18]. However, this is not always the case, as venous blood currents result in the flow of eggs in the opposite direction, preventing environmental escape and causing entrapment [19]. For endogenous populations, the clinical and epidemiological picture of this disease is quite different. Chronic schistosomiasis is well-established among endemic residents, whose intensity of infection depends on continuous exposure to infested waters and the individual’s immune response to infection, resulting in either the intestinal or urinary form of the disease. Infection may be acquired from as early as five years and below, peaking at 8–15 years of age and persisting at moderate levels throughout adulthood [20,21]. Intestinal schistosomiasis results in sporadic abdominal pain, per-rectal bleeding, chronic diarrhea caused by ulceration, mucosal hyperplasia, abscess formation, and polyposis due to the formation of granuloma along the intestinal wall and liver [22]. Poor regulation of the immune response causes chronic patients to subsequently develop hepatosplenic periportal fibrosis, commonly known as Symmer’s pipe stem fibrosis, where scarring results in enlarged liver veins that have the appearance of pipe-stems in cross-section. This affects hepatocellular function and results in portal hypertension and splenomegaly [15,21]. Urinary schistosomiasis emerges as a consequence of lesions present on the bladder wall, resulting in bloody urine (hematuria), chronic pain, anemia, pollakisuria, proteinuria, and dysuria [20]. Hematuria is the most common and widespread lower tract symptom that is often regarded as a sign of puberty in young boys and girls of many endemic regions. Bladder malignances, calcification, and fibrosis of the urinary tract resulting in obstructive uropathy are also observed [22]. Further complications include the development of inflammatory lesions in female genitalia, resulting in sandy patches that not only cause pain but also increase infertility rates, spontaneous abortion, and risk of HIV transmission [15]. Growing evidence additionally implicates *S. haematobium* as a Group 1 definitive biological precursor to bladder cancer occurrences [23]. It has been suggested that schistosome eggs release vascular endothelial growth factor (VEGF) proteins that contribute to the incidence of bladder cancer. Inflammation induced by *S. haematobium* infection has been associated with the activation of carcinogenic metabolites produced by aromatic amines and polycyclic aromatic hydrocarbons [24]. 

## 3. Effector Cell Activation during Schistosome Infection

As is evident, one of the most notable characteristics of schistosome infection is the down-regulation of the host’s immune response to infection. The schistosome’s rare ability to reside within the host for decades suggests that it is able to modulate the immune response to evade elimination, which is in itself surprisingly beneficial to the host through the modulation of over-vigorous immunopathology, thereby disrupting severe disease. During infection, eggs produced by the female parasite are both immunogenically and metabolically active. Eggs retained in the host give rise to cell-mediated immunopathological reactions that in turn cause the formation of granuloma that comprise various T-cell dependent responses, most of which are Th2-related [25]. Heterogenous metabolic secretions in the form of antigenic glycoproteins and glycolipids released from live eggs are responsible for this phenomenon. These are responsible for egg transit and in the process, promote schistosomiasis transmission. These soluble egg antigens (SEA) are able to provoke Th2 responses in naïve mice, as well as other immunologic reactions that include the production of various inflammatory or innate cells such as mast cells, dendritic cells (DCs) and eosinophils, which accumulate around the eggs as granulomas [26,27].

### 3.1. Dendritic Cells 

One of the most important antigen-presenting cells (APCs) involved in regulating the immune response are precursor and immature DCs, which travel from the blood into peripheral tissues and regulate foreign antigens by internalizing them for presentation to T cells in the lymph nodes [28]. Maturation of these sentinel cells, stimulated by IFN-γ, CD40L, TNF-α or pathogen-derived signals, leads to the activation of naive Th cells, resulting in proliferation and differentiation into Th1, Th2, or T regulatory cells, which in turn drive the development of specific forms of immunity. These distinct DC phenotypes are, however, highly dependent on the state or condition of the sentinel DCs in the presence of released mediators from provoked tissues. Tissue-derived cytokines such as prostaglandin E_2_ (PGE2) or IFN-γ during activation generate Type 1 effector DCs (DC1s), which in turn produce high levels of IL-12 when interacting with naïve T cells, while the development of Th2 cells is driven by IL-12-deficient Type 2 DCs (DC2s) (Figure 2) [29]. A study by de Jong and colleagues demonstrated that microbial compounds induce Th polarization. Monocyte-derived DCs cultured in the presence of SEA showed DC maturation and repolarization resistance, indicating that some kind of protective immunity is developed through the polarization of the DC1 and DC2 subsets [29]. 

For recognition, DCs express receptors in the form of C-type lectins (CLRs) and Toll-like receptors (TLRs) that easily recognize self or non-self glycoprotein carbohydrates, a component abundantly present in SEAs [28]. The response of cytokines can partly be controlled by glycosylated antigens in SEA and these can be recognized by molecules released by the host through APCs called pattern recognition receptors (PRRs). These are responsible for distinguishing between various groups of pathogens and the receptors that bind to molecules released from tissues, either during infection or constitutively, as well as characterizing the type and response of pathogen-specific responses of tissues. Of most importance are CLRs, which include dectin 1/2, DC-specific ICAM-3-grabbing nonintegrin (DC-SIGN, CD209), mannose receptor (MR, CD206) and DC immunoreceptor (DCIR, CLECHA, CD367) [30]. DC-SIGN and macrophage galactose-type lectin have previously been shown to have the ability to bind to SEA strongly, while the two in conjunction with MR are involved in both the binding and internalization of SEA, leading to the maturation and modulation of DC function and their subsequent T cell polarizing abilities [28,31]. Kuipers and coworkers later showed that DC-SIGN and MR have the ability to recognize Galβ1-4(Fucα1-3)GlcNAc, also known as LewisX (LeX or Le^X^), an immunogenic glycan-motif present in schistosome egg E/S and larvae. Extracellular vesicles are internalized by human monocyte-derived DCs (moDCs) via DC-SIGN owing to the presence of DC-SIGN signals such as LeX, which increases the expression of cytokines such as IL-12 and IL-10 [30]. The production of the IL-10 cytokine has been reviewed severally as one that is pivotal to schistosome disease progression, as it is responsible for down-regulating the Th1 response, but at the same time preventing severe pulmonary disease in chronic schistosomiasis via a mechanism that is independent of the Th2 response by regulating hepatic humoral immunity (Figure 1) [32]. Th12, on the other hand, is associated with high levels of IFN-γ during the Th1 response through its ability to stimulate Th1 and Natural Killer cells for its production [33]. 

Omega-1, a glycoprotein secreted by live schistosome eggs that is also present in SEA, actually initiates and conditions moDCs to exhibit Th2-polarized responses from naive CD4+ T cells in vitro in ways that closely resemble whole SEA. This was further confirmed in IL-4 dual receptor mice, which when treated with both recombinant and natural omega-1 alone, generated a Th2 response in vivo, while omega-1-depleted SEA failed to produce the same response in vitro but not in vivo [34]. This was substantiated by Steinfelder et al. who showed that omega-1, also referred to as T2 ribonuclease, has a direct influence on the morphology and ability of DCs to have physical interaction with CD4+ T lymphocytes by either strengthening or lowering the activation signals delivered by DCs [35]. Once omega-1 is internalized, protein synthesis is reduced via RNA breakdown, resulting in the anti-inflammatory DC phenotype [36]. 

### 3.2. Eosinophils

It is unquestionable that eosinophils play a vital role in schistosomiasis. They are regarded as multifunctional leukocytes with both effector and immunomodulatory functions and are modulators of innate and adaptive immunity in their own right by initiating and intensifying various inflammatory responses [37]. In most cases, an increase in eosinophil count and the presence of a dry cough indicates the onset of acute schistosomiasis or a Th1 reaction due to larvae migrating to various tissues [17]. Their prominence may also be due to the release of antigens either caused by spontaneous parasitic death or chemotherapy treatment [38]. Eosinophils are made out of granule proteins whose crystalloid consists of major basic protein (MBP), while the matrix contains eosinophil peroxidase, eosinophil cationic protein (ECP) and eosinophil-derived neurotoxin, which are responsible for the cytolytic and inflammatory properties exhibited by eosinophils through the production of cytokines such as TGF-alpha, IL-1, IL-3, and GM-CSF [39]. Eosinophils, however, display dual activity by providing protective immunity during infection but also contributing to pathology, especially in association with allergic reactions [40]. Through the antibody-dependent cellular cytotoxicity mechanism, eosinophils have the ability to kill parasite larvae through recognition of glycan epitopes on the glycocalyx of cercariae and schistosomulae [40,41]. Adult schistosomes escape this mechanism by concealing their outer tegument from recognition using host antigens such as tissue- and blood-type antigens, immunoglobins, β2 microglobulin and complement components [38]. This shield is ultimately broken down by eosinophils as the worm dies, whether naturally or from chemotherapeutic treatment. The death of larval worms, on the other hand, occurs as a result of effector cells that include eosinophils, macrophages, NK cells, and neutrophils interacting with antiserum from hosts infected with schistosomiasis. The IgG antibody contains the variable fragment region that binds to the schistosome parasite and the constant fragment (Fc) domain that causes the IgG antibody to interact with the Fc receptors of various effector cells (Figure 2). The combination of these interactions results in the activation of effector cells to release cytotoxic molecules such as perforins, granzymes, and eosinophils, which kill schistosome parasites [41,42]. In addition, in vivo experiments have shown the killing of worms in both rats and humans by eosinophils, platelets, and mononuclear phagocytes in the presence of certain anti-schistosome IgE antibodies [41].

Apart from this, a few other studies have observed the potential role of eosinophils in host protection against schistosomiasis. A longitudinal investigation was carried out by Ganley-Leal and co-workers on adult male car washers who through their occupation were relatively exposed to schistosome infection from Lake Victoria. In addition, a third of the cohorts were HIV-1-seropositive. The study reported increased peripheral eosinophilia, which was associated with resistance to *S. mansoni* re-infection in HIV-1-seronegative persons, while reduced eosinophil percentages correlated with low CD4-T-cell counts in HIV-1-seropositive cohorts. This implicates the role of these immune cells in not only schistosome host protection, but also increased susceptibility to re-infection caused by HIV-1 co-infection [43]. However, the unequivocal way in which eosinophils provide partial immunity to the host is still not well understood and requires more research.

As mentioned earlier, eosinophils through various mechanisms are also a large contributor to host tissue damage and pathology, which may be physical damage, direct cellular cytotoxicity or thromboembolic effects due to eosinophil-induced hypercoagulability. However, a large part of tissue damage is caused by the release of eosinophil cationic proteins (granule proteins), cytotoxic compounds and other toxic inflammatory mediators such as reactive oxygen species, platelet-activating factors, lysosomal hydrolases, and leukotrienes [44]. Most notable is the release of the highly positively charged cationic MBP and ECP [45]. All these substances are not only toxic to both the schistosome parasites and the cell in vivo and in vitro, but also affect host organs such as the heart or brain (ROS and hydrolases) and play a role in the human immune response (leukotrienes and platelet-activating factors) [17,44].

### 3.3. Macrophages

Apart from eosinophils, macrophages are essential innate cells that play various roles in host protective immunity, immune regulation, schistosome-induced inflammation, and fibrosis, as well as wound healing [46]. Their ability to display such a broad range of activities from promoting and restricting to repairing damage via various mechanisms is highly dependent on their activation. During the Th1 response, IFN-γ, which regulates the expression of antigens and major histocompatibility complex (MHC) class I and II proteins, partners with microbial products to “classically” activate macrophages (CAMϕs, also known as M1 macrophages) to elicit a microbicidal response that causes damage to surrounding tissue, while Th2 responses use IL-13 and IL-4 cytokines to “alternatively” activate macrophages (Figure 2) (AAMϕs, also called M2 macrophages) [46]. Both opposite and cross-regulatory effects are displayed by macrophages through the production of IL-10, and IL-12, whose cytokines interact with Th1 and Th2 cytokines in a complex interregulatory network. IL-10, in conjunction with IL-6, expresses proinflammatory cytokines such TNF-α, which can also stimulate the production of IL-10 and IL-1 in response to IFN-γ, which is produced by IL-12 [47]. Schistosomal eggs are potent antigenic focal points for the human immune response, mostly in the form of granulomas that are composed of immune cells, macrophages included. The polarization of macrophages to induce an AA phenotype is triggered by Type 2 cytokines, which induce inflammation and tissue damage until a maximal response is attained. Thereafter IL-13, using AA macrophages and fibroblasts, causes granulomas to decrease in size and become fibrotic [48]. The activation of AA macrophages is characterized by signature genes such as resistin-like beta (*Relmα*, Retnla or Fizz1), arginase 1 (*Arg1*) and Ym1 (*Chil3*), resulting in the prevention of pathology through the formation of granulomas, facilitation of tissue regeneration, and repair, as well as the regulation of T-cell proliferation. Worth noting is their ability to inhibit the differentiation of M1, which are mostly activated by IFN-γ and TLR agonists, making them responsible for causing oxidative tissue damage through the expression of nitric oxide (NO) synthase (iNOS and *Nos2*) and the generation of pro-inflammatory cytokines [48,49].

### 3.4. Mast Cells

The recruitment of mast cells (MCs) is an important feature and hallmark during most helminth infections, schistosomiasis included, and these cells are activated and degranulated by the cross-linking of immunoglobin E (IgE) via the FcεRI receptor (also called the high-affinity receptor) [50]. The interaction of these two factors causes MCs to activate and produce cytosolic granules such as TNF, proteases, growth factors, and cytokines such as IL-4 and IL-5, MC protease 1 (MCPT-1), and VEGF (Figure 2) [51]. In addition, MCs are also activated and proliferated by interactions between the ST2 receptor and IL-33. Once this process takes place, MCs undergo degranulation to release mediators such as histamine, IL-4 and IL-13 that are involved in activating alternative macrophages (M2), TNF-α, CXCL1, and CXCL2, resulting in neutrophil accumulation at the site of infection, and prostaglandin D2 causing IL-33 cleavage, which in turn increases the induction of Type 2 innate lymphoid cells via the CRTH2 receptor. On the other hand, MC degranulation is also associated with host protection by preventing worm development or encouraging the expulsion of parasites by altering the tissue environment into a toxic one that is based on increasing the production of mucin, inducing goblet cell hyperplasia and contraction of intestinal smooth muscle. It has also been shown in mice that the accumulation and proliferation of MCs (mastocytosis) are not only time- and location-dependent, but also coincide with motility-related gastrointestinal effects and the production of eggs [29].

### 3.5. Basophils

Basophils, sometimes referred to as “redundant MCs” or “poor MC sisters”, share a few characteristics with MCs and other innate immune cells such as eosinophils because of the way they originate from the CD34^+^ progenitor cell [52]. Although they are quantitatively the least of all granulocytes in the blood, bone marrow, and spleen, there is growing evidence that these cells do play an important role in immunoregulation. Schistosome eggs are the main drivers of basophil induction whose expression up-regulates the production of IL-4, which is highly involved in the Th2 response, as well as VCAM-1 expression, enhancing leukocyte recruitment to affected tissues [53]. All this suggests the role of basophils in schistosome inflammation and pathology. Apart from that, basophils are associated with other helminth infections and chronic allergic inflammation, while a decrease in their population results in a decrease in egg granuloma development [54].

Data produced by Fraga and co-workers suggest that eggs produced 5–6 weeks after natural schistosome infection are exposed to an immunologic milieu, where basophils and CD4+ T cells are both primed for IL-4 production in response to schistosome antigens, further implicating these cells in the initiation of the Type 2 response [55]. A potent inducer for the release of IL-4 and IL-13 from basophils is the IPSE/alpha-1 immunoglobin-binding factor that interacts at very high affinity with IgE bound to the FcεRI receptor, which is also found on the surface of basophils irrespective of IgE antigen specificity (Figure 2) [56,57]. It has been shown that IgE stripping abrogates basophil activation, while resensitization restores the IPSE/alpha-1 cytokine-releasing ability on basophils [58]. Other studies have also shown that unstimulated basophil cells or IPSE/alpha-1 depleted SmEA do not produce IL-4, while the exact opposite was observed in anti-IgE basophils, SmEA, and both unglycosylated and glycosylated recombinant IPSE/alpha-1 [59,60]. It is of particular and utmost importance for the survival of the host that IL-4 and the subsequent Th2 response be induced. Brunet et al. were able to show that IL-4 prevents severe schistosomiasis by comparing disease progression in IL-4 deficient and wild type mice. The results revealed that the former experienced acute cachexia and eventually died, while the wild type C57BL/6 mice developed chronic schistosomiasis and were able to tolerate infection [61]. The absence of the IL-4Rα chain in macrophages also results in death [60]. Once activated, basophils release several mediators that include histamine, proteolytic enzymes, proteoglycans, several other cytokines, and leukotrienes [54].

### 3.6. Breg and Treg cells

Regulatory immune cells not only contribute to disease progression, but also play crucial roles in host protection, and the development of regulatory B (Breg) cells in both men and mice forms part of this network [62]. Through the expression of IL-10 and stimulation of T (Treg) cells, B cells gain the ability to block pro-inflammatory immune responses. This is achieved by eggs and/or egg E/S acting directly on splenic B cells, which bind and internalize the egg antigens on pattern recognition receptors (PRRs) such as TLRs. This in turn drives IL-10 secretion, which subsequently stimulates Treg cell development. A report by Fairfax and colleagues demonstrated the development of severe disease in the form of portal hypertension following IL-10R blockage [32]. It was further discovered that IL-10 actually regulates CXCL9 and CXCL16 chemokine expression, which is responsible for recruiting B cells into the liver, thereby regulating inflammation within the organ. It was observed in Gabonese *S. haematobium*-infected adults that CD1d^hi^ and CD24^hi^ CD27^+^ B cells increased IL-10 and latency-associated peptide levels, the latter being a component of the TGF-β complex, while reduced T-cell cytokine responses, but enhanced Treg cells were observed in B cell-T cell co-cultures [63]. B cells are also potent producers of IgE, which exerts its functions through its FcεRI high affinity or FcεRII or CD23 low affinity receptors. CD23 is also directly produced by B cells, whose expression can be enhanced by IL-4. It has been implicated in the active development of resistance to *Schistosoma* re-infection [64,65].

B cells make no contribution to granuloma formation, as seen in studies using B-cell-deficient mice, but are required for Th2 T cell responses [66,67]. This effect is opposite in *S. japonicum* infections though, as the B cell is required for granuloma formation. This was observed when B cell-impaired mice developed smaller granulomas five weeks post-infection compared to the controls, but later showed no significant difference in granuloma pathology eight weeks subsequent to infection [66]. T cell activation, on the other hand, is crucial for granuloma formation and prolonged survival of the host. The expression of co-stimulatory molecules such as CD69, CD80, and CD86 rapidly up-regulates following infection and recognize and bind to ligands present on APCs [68,69]. The cytokine environment to which APCs are exposed gives rise to effector T cell differentiation in the form of Th1, Th2, and CD4^+^CD25^+^, also known as Treg, phenotypes. Other subsets of cells include Th9, Th17, Th22, and follicular T cells [70]. The function of the IL-10 cytokine is again essential to immune cell growth and the immune response, which it controls by down-regulating MHC class II, CD40, CD80, and CD86 [71]. It is a well-known fact that the Th1 and Th2 responses are associated with respectively early infection and chronic disease during egg production. Treg cells play a role in modulating granuloma development by accumulating in them and limiting some immune functions through the production of IL-10, which causes an increase in Th1, Th2, Th17, IgG1, IgG2b, IgE, and eosinophilia levels [69,72]. The expression of these cytokines in turn blocks subsequent development of inflammatory cytokines. Treg cells are similar to Bregs in that the latter are also involved in inflammatory suppression through the secretion of IL-10. It has moreover been shown that their transfer causes Treg cell recruitment to inflammatory airways in an IL-10-dependent approach [73].

In schistosomiasis, the presence of the above-mentioned inflammatory cells gives rise to the distinctive egg-induced granuloma, which is detrimental to the host because of developing fibrosis and portal hypertension, but simultaneously helps the host survive with infection for several years [74]. It does so by providing a potent shield between toxic egg secretions and the host’s tissues. Interaction between these has proven fatal, as has been shown in immunocompromised mouse models, thus it presents itself as a compromise for infection.

### 3.7. MicroRNAs

Although not effector cells, miRNAs play a crucial role in effector cell and immune responses during schistosome infection. miRNAs are non-coding RNAs that are transcribed from DNA and are very important regulators of gene expression due to their involvement in mRNA degradation and repression or activation of translational and transcription regulation. Due to their importance in normal animal development as well their secretion into extracellular fluid, the abnormal expression of these exosomes is associated with human disease and may well be used as potential disease biomarkers respectively [75]. miRNAs have been shown to play a role in schistosome survival. Extracellular vesicles released from adult *S. japonicum* parasites (SjEVs) regulate TNF-α production and cell proliferation via miRNAs, thereby increasing host monocytes and contributing to schistosome survival [76]. SjEVs are engulfed by monocytes that eventually develop into macrophages which then play a role in inflammation. Therefore, the regulation of TNF-α by miRNAs indirectly or directly alters macrophage immune cell function. In addition, studies have also shown that SjEVs also plays a role in TLR and TNF signal pathways by increasing the expression of molecules that are involved in these pathways.

Differential expression of over 100 miRNAs have been observed in *S. japonicum* models, both before and after infection and are involved in the immune response, apoptosis and signaling pathways, cell differentiation, and nutrient metabolism [77]. Many studies have illustrated the importance of miRNAs in the regulation of both innate and adaptive immunity responses within murine models and regulating signaling by Toll-like receptors and cytokines as well as regulating signaling by T-cell receptors and antigen presentation, respectively. The deregulation of miRNAs has been observed during hepatic fibrosis progression in *S. japonicum* infected mice with the inhibition of certain miRNAs (miR-21 and miR-351) providing protection to the host against fatal schistosomiasis [78]. Consequent research has also suggested the role of miRNAs in regulating hepatic fibrosis by activating HSCs via the downregulation of miR-203 thus increasing IL-33 expression, which stimulates IL-13 and consequent production of hepatic lymphoid cells. miRNAs such as miR-21 are elevated during hepatic fibrosis and studies that have shown the reduction of the latter due to the inhibition of the molecule by adeno-associated virus serotype 8 (rAAV8), which inhibits IL-13/SMAD and TGF-β1/Smad signaling pathways. Therefore, miR-21 has been proposed as a novel target for the potential treatment of schistosome-induced hepatic fibrosis along with miR-454, whose expression has also been detected in the liver of schistosome-infected mice, which by targeting Smad4 is able to deactivate hepatic stellate cells and TGF-β1-induced LX-2 cells [77].

## 4. Schistosome Eggs and Paradox of the Granuloma

Schistosome adults live together as permanently embraced couples, with the long and thin female lying within the male’s gynaecophoric groove and about 28 days following copulation, they lay hundreds to thousands of eggs daily onto the endothelial lining of capillary walls, which are distributed via blood flow through intestinal epithelia to the lumen for urinal or fecal excretion [79,80]. Unfortunately, about 50% of these eggs are transported via portal circulation within the bloodstream to the liver (*S. mansoni*), where they get caught up with no escape route, because the diameter of the sinusoids are too small for the eggs to pass through [81,82]. The eggs are highly antigenic, harmful, and metabolically active organisms [83]. The eggshell is particularly important, as it interacts directly with the host’s immune system. Vitelline cells composed of eggshell precursor proteins envelop the oocyte once it becomes fertilized within the female reproductive tract. The precursor eggshell proteins become cross-linked by quinine tanning due to tyrosinase activity between both single and neighboring proteins, causing them to harden and become protease-resistant but porous at the same time to enable free movement of egg secretion products (ESP) [84]. Tyrosinases are glycoenzymes that contain copper and catalyze the conversion of _L_-DOPA to *o*-quinones through two subsequent processes: hydroxylation of tyrosine residues to _L_-DOPA (_L_-dihydroxyphenylalanine) and the oxidation of _L_-DOPA to *o*-quinones (*ortho*-quinones) [84,85]. *O*-quinones are extremely reactive and become even more so by reacting with nucleophilic compounds (such as sulfhydryl groups and amino acids) on neighboring proteins to produce adducts. Among the major eggshell proteins, which include p14, p19, and p48, are glycoproteins such as P40, thioredoxin peroxidase and phosphoenolpyruvate carboxykinase, which are antigenic and initiate cellular and antibody responses. Once the eggs mature after seven days, an inner envelope is formed just beneath the shell that is extremely metabolically active and is considered the central source of the egg’s highly immunogenic secretions. These are composed of IPSE/alpha-1, Kappa5, and Omega-1, compounds that are all present within whole eggs, ESP and SEA, and are major drivers of the Th2 response [84,86]. In addition, the eggshell is also responsible for encapsulating the miracidia, which although it does not hatch within the host, is a potent source for the release of SEA [87]. Both the deposition of eggs and SEA promote fibrinolytic and endothelial activity as well as angiogenesis, which in turn induces TNFα, IL-1, and chemokine production that provokes extravasation in the form of recruited monocytes, neutrophils and lymphocytes [87,88].

One of the most obvious and distinct clinical features and hallmarks of schistosome infection is the formation of the granuloma, a collective mass of collagen-bound inflammatory cells that encapsulates mature parasite eggs with a wall during the chronic phase of infection and is responsible for the mortality and morbidity demonstrated in infected humans. The granuloma cell population is in general a large aggregation of mononuclear phagocytes, plasma cells, neutrophiles, fibroblasts, and lymphocytes, but is typically composed of 50% eosinophilia, 30% macrophage, and 20% CD4^+^ T cells (Figure 3) [89,90]. Formation of the granuloma is a process principally driven by cell-mediated immune responses orchestrated by T helper lymphocytes that exhibit a CD4^+^ phenotype (Th1 and Th2 cells) and on a lesser note CD8^+^ T cells, B cells, and M2 macrophages [32,83]. The Type 1 response induced after infection persists for ~5 weeks and is characterized by IL-12 and IFN-γ, while the Th2 shift experienced ~5–6 weeks post-infection, following the production of eggs, results in the decrease of IFN-γ [80]. Th2 cells and macrophages then become polarized and the Th2 response is typified by the intense presence of eosinophils and basophils, increased IL-4, IL-5, and IL-13 production, as well as an IgGI to IgE isotopic switch. Individuals infected with *S. mansoni* develop severe hepatosplenic schistosomiasis, which is characterized by ascites, portal hypertension, terminal bleeding and obstructive vascular lesions caused by liver portal tract fibrosis [83]. For the most part, the pathological conditions displayed during chronic disease are largely caused by the host’s reactions to eggs confined within various tissues, mostly the liver, and egg secretions that polarize the environment to a Th2 immune response that evokes inflammation in the form of granulomas, which in turn cause fibrosis [91]. Formation of the granuloma occurs around individual eggs, which once they die results in healing and subsequent fibrotic plaques (Figure 3). The principal cytokine responsible for this is IL-13 and studies have shown failure to develop severe hepatic fibrosis in IL-13 knockout (KO), ineffective or neutralized mice [92]. The release of antigens from schistosomal eggs activates the production of IL-33 from injured hepatocytes and IL-4 from innate lymphoid cells. The activity of IL-4 and IL-33 on naïve T cells promotes Th2 differentiation and in turn the release of large amounts of IL-4, IL-5, and IL-13 cytokines [93]. IL-4 and IL-13 activate AA macrophages through the expression of arginase1, Relmα and Ym1 signature genes that potentially inhibit fatal host pathology. Their involvement includes the formation of granulomas by recruitment of collagen and immune cells, tissue repair and regeneration, inhibition of CAMs and regulation of T cell proliferation [48].

The role of IL-13 is particularly relevant because its activation causes the expression of arginase, which cleaves L-arginine to form L-ornithine that is ultimately converted to proline (via ornithine aminotransferase), an important component in collagen production and the development of fibrosis [88]. In addition, IL-13 activates HSC differentiation, which is another potent source of collagen and is also involved in fibrogenesis [88,93]. In *S. japonicum* infections, HSC proliferation by SEA induces the secretion of collagen I and III, which cause damage to live tissues [94]. Cytokine profiles in other studies have also shown an increased progression of IL-5 and IL-13 expression among patients experiencing severe hepatic fibrosis over an extended period of one year without treatment [95].

Although IL-13 is the dominant pro-fibrotic mediator in hepatic schistosomiasis, the role IL-4 and its receptors play cannot be ignored. Studies have shown decreased mRNA expression of IL-5 and IL-13 Th2 cytokines and increased mRNA expression of IFN-γ and IL-2 Th1 cytokines in anti-IL-4 treated infected liver. Experiments conducted in IL-4/IL-4 receptor KO mice show that such an increase results in reduced hepatic granulomas and fibrosis, demonstrating that the IL-4 receptor is actually more central to the formation of granulomas [96] The IL-4-IL-13 receptor complex comprises three combinations, the more popular being the Type II receptor that consists of the IL-4Rα and IL-13Rα1, with both IL-4 and IL-13 employing the IL-4Rα chain [97]. Phosphorylation of the signal transducer and activator of transcription factor 6 occurs once IL-4 and IL-13 bind to the IL-4Rα subunit, which sequentially activates downstream signaling that regulates factors involved in fibrosis, inflammation, and vascular remodeling [6]. A study conducted by Fallon and co-workers [98] also reiterates the significance of these cytokines cooperating for granuloma formation on *S. mansoni*-infected mice. IL-4/IL-13 double KO mice showed reduced liver granulomas and improved fibrosis along with IL-13 KO mice [96]. IL-4 KO mice, on the other hand, exhibited no significant reduction on collagen accumulation or liver fibrosis.

The generation of the granuloma is also largely driven by SEA, which are not only highly glycosylated but also hold immunogenic consequences. The production of eggs is an important indicator of the commencement of chronic schistosomiasis caused by SEA and their carbohydrate moieties, which include GalNAc(b1-4)[Fuc(a1-3)]GlcNAc (LDN-F), Fuc(a1-3)GalNAc(b1-4)GlcNAc (F-LDN), and Fuc(a1-2)Fuc(a1-3)GalNAc(b1-4)[Fuc(a1-2)Fuc(a1-3)]GlcNAc (DF-LDN-DF) [99]. These egg components are actively involved in polarizing the host’s immune response from the Th1 to the Th2 type response. Antibodies that are generated in response to egg antigens result in the formation of the granuloma (which in turn assists the passage of eggs to the intestinal lumen for excretion), and the recruitment of inflammatory cells mostly in the form of T cells, eosinophils and macrophages [100]. Apart from the above-mentioned glycoconjugates, Gal(b1-4)[Fuc(a1-3)]GlcNAc, commonly referred to as Lewis X or the Le^x^ antigen, is a chief target of the host immune response. Glycans that contain the Le^x^ determinant have the ability to trigger the IL-10 and prostaglandin E2 Th2-associated mediators via the proliferation of B-cells, as well as induce IL-10 secretion by stimulating peripheral blood mononuclear cells via the TLR4 receptor. Examples of such Le^x^-containing glycans include the IPSE/alpha-1, which also triggers IL-4 production by stimulating basophils, and the lacto-N-fucopentaose (LNFPIII) fucosylated glycans [99,100].

Because of the presence of parasite eggs, metabolic modifications and physical impairment may occur in the host, partially caused by the involvement of ROS produced by resulting granulomas [101]. In their defense, ROS are actually generated primarily for the destruction of eggs but eventually become part and parcel of the pathology associated with schistosomiasis. Eosinophils recruited by the schistosome-induced granuloma generate oxygen free radicals in the form of superoxide and hydroxyl radicals that in turn discharge active eosinophil peroxidase, which surrounds and circulates around the granuloma [102].

## 5. Reactive Species Production in Schistosomes

The vertebrate human host operates as a potent source of oxidants. Professional phagocytes are cells of the innate immune system capable of producing large amounts of bactericidal ROS, which coupled with redox-active and toxic by-products, factor into the schistosome’s resistance against oxidative stress [103]. Through phagocytosis, also referred to as oxidative or respiratory burst, phagocytic cells (leukocytes) form a cytotoxic defense against helminths, parasitic fungi, mycobacteria, and protozoa [104]. Phagosomal membrane-bound nicotinamide adenine dinucleotide phosphate (NADPH) oxidase enzyme is activated, thereby reacting and consuming molecular oxygen to produce superoxide anion (O_2_^−^). The superoxide anion forms the basis for additional ROS generation in the form of hydrogen peroxide (H_2_O_2_), a weak oxidizing agent, through the conversion of O_2_^−^ by superoxide dismutase (SOD) [105]. Through the Fenton reaction, Fe^2+^ reacts with H_2_O_2_ to form the highly reactive hydroxyl radical (OH**^∙^**), hydroxyl ion (OH^−^), and ferric iron Fe^3+^. The last-named will in turn react with O_2_^−^ to be reduced back into Fe^2+^, allowing for its conversion to H_2_O_2_ and OH**^∙^** again (Haber-Weiss pathway) [106]. Phagosome cytotoxicity is increased through phagosome fusion caused by hypohalous acids such as hypochloride (OCl^−^) derived from H_2_O_2_. OCl^−^ may additionally react with H_2_O_2_ and amine, producing singlet oxygen (^1^O_2_), which in turn forms the basis for other free radicals. Superoxide may also react with NO to produce peroxynitrite, which may further react with carbonyl compounds to form organic peroxyl radicals (Figure 4) [105]. Through the stimulation of cytokines and other immunological reactions, macrophage NOS in the form of NOS2, macNOS, and inducive NOS (iNOS), which is in fact never produced within resting cells, is produced [106]. In schistosomiasis cases, iNOS is expressed primarily by macrophages and most notably in response to IFN-γ and other pro-inflammatory cytokines such as TNF-α and IL-1β [47,107]. It has acted as a host protective molecule by regulating inflammatory responses and mediating the cytotoxic pathway responsible for killing 2-week-old schistosomulae in vitro by activated macrophages [107,108]. Despite its beneficial roles NO has been found to stimulate disease processes by forming toxic radicals that induce cell apoptosis and damage. NO is oxidized to nitrogen dioxide (NO_2_) or may generate peroxynitrite as explained above, which in turn induces lipid peroxidation and DNA damage by oxidizing sulfhydril groups and decomposing OH**^∙^** radicals and NO_2_, respectively [109].

Other implications of ROS produced by immune cells include NADPH oxidase released by phagocytes. The O_2_ in reactive oxygen is transformed in a bid to attack microorganisms. This however, results in the accumulation of inflammatory cells such as macrophages and lymphocytes, which leads to the formation of the granuloma [9]. Thus, the production of ROS promotes schistosome infection progression. A study conducted by El-Sokkary and co-workers discovered and confirmed that the production of free radicals is actually an underlying factor in disease progression and the pathological changes that are associated with schistosomiasis. Oxidative processes in schistosome-infected mice were investigated using the antioxidant melatonin. Untreated mice observed an increase in lipid peroxidation and NO levels, while SOD activity, glutathione, and vitamin E levels in the spleen, kidney, and liver saw a decrease in levels. Granulomas were present in the liver with an associated presence of eosinophils as well as the manifestation of necrotic cells and megakoryocytes in the kidney and spleen of the mice respectively [110]. Similar studies conducted by Gharib and colleagues showed that the presence of parasitic eggs stimulated endogenous eosinophil peroxidase release, while a decrease in the activities of glutathione peroxidase, superoxide dismutase as well as catalase was observed [111].

The schistosome worm depends on the human host for survival, development, and reproduction by consuming red blood cells (RBCs) from which host Hb is catabolized. Ingestion of RBCs is essential for the parasite’s nutrition, especially for amino acids, peptides, fatty acids, triglycerides, amino sugars, cholesterol, iron, and calcium [112]. The consumption of Hb, however, leads to the production of dipeptides or free amino acids that are taken up by the parasite’s cecum and are broken down by proteolytic enzymes to be eventually regurgitated into the bloodstream as toxic heme and cellular debris [48,113,114]. Accumulation of these substances, particularly in the liver, not only induces oxidative stress and inhibits mechanisms set in place to defend against ROS and RNS, but are also consumed by macrophages that induce the expression of the RELM-α Th2 negative regulator [48,113]. Toxic heme, also known as free iron (II)-heme, is referred to as such owing to its ability to oxidize parasite biomolecules and reduce molecular oxygen, which is paramount in the production of ROS [114]. Reduction of molecular oxygen subsequently generates redox reactive substances, which are directly involved in lipid peroxidation. This is aggravated by reactions between the active heme, lipids, and intracellular H_2_O_2_ [115]. Autoxidation of Fe^2+^, via the Fenton (iron-catalyzed Haber-Weiss) reaction, leads to the production of non-functional ferric heme (Fe^3+^) and O_2_^−^ that in turn generates H_2_O_2_, which also has the potential to react with Fe^3+^ to produce the ferryl heme (Fe^4+^) intermediate. These products, as well as their intermediates, are extremely toxic and are known to oxidize lipids, amino acids, and nucleic acids, as well as produce toxic heme that ultimately leads to protein-cross-linked species, actual loss of heme and the release of free iron. In addition, “free” heme also induces cellular damage by disrupting the solubility and stability of phospholipid membranes because of its amphyphilic nature [116].

To counter the cytotoxic effects associated with toxic heme, hematophagus organisms, such as malarial *Plasmodium* parasites, the *Rhodnius prolixus* kissing bug and schistosomes, employ a detoxifying mechanism to either detoxify or trap the released heme [117]. The free heme, also known as ferriprotoporphyrin IX (Fe(III)PPIX), is assembled into a biomineral made up of centrosymmetric (head-to-tail) dimers consisting of two covalently-linked heme molecules through iron-carboxylate coordinate bonding. These dimers then form the crystal lattice of hemozoin (Hz) when they hydrogen-bond to other dimmers [115,117]. The core of the crystal (which consists of Fe^3+^) is closed off and is therefore unable to take part in any oxidizing reactions, thus making Hz less toxic to the parasite. It is for this reason that this dark brown pigment accounts for up to 50% of all heme content in female schistosomes [116,117].

## 6. Implications of ROS in Genetic Instability

Oxidative stress in schistosomiasis is accompanied by mediated site-specific tissue damage such as in the liver, especially at sites of granulomatous inflammation, owing to the production of ROS by macrophages and eosinophils [118]. Lipid peroxidation occurs, damaging antioxidant systems set in place for ROS scavenging; this in turn results in peroxidative impairment of lipids found in the liver [119]. Lipid peroxidation is a complex reaction that that has been associated not only with disease progression but also with the inflammatory process, which involves the oxidation of lipids and polyunsaturated fatty acids to produce malondialdehyde, lipid hydroperoxides, and cyclic peroxides. Such products generated via lipid peroxidation, which are extremely genotoxic and mutagenic, react with DNA and induce damage [109]. Oxidative stress in general stimulates pathways that damage cell structure and function, thus linking it to the initiation and progression of cancer by inducing DNA mutations and damage and causing cell proliferation and genome instability [120]. *S. haematobium* has been mentioned frequently to be classified as a Group 1 carcinogen by the World Health Organization and the International Agency for Research on Cancer in response to reports on the disease predisposing infected individuals to carcinomas such as squamous bladder cancer [20]. ROS released by inflammatory cells causes DNA-strand breaks, sister chromatid exchanges, as well as mutations. It has been shown that schistosomal infection and subsequent inflammations result in the generation of free radicals, which become oxidized, causing single-strand breaks in DNA. Because of this complication, 8-hydroxy-2′-deoxyguanosine (also known as 8-OHdG), which is a well-known biomarker of oxidative stress, increases owing to the presence of DNA-repair genes such as deoxy8-oxyguanine-DNA-glycosylase as well as apurinic/apyrimidinic endonuclease [121]. Other studies have also indicated that nitrative stress generated by NO forms 8-nitroguanine [122]. Another notable way in which oxidative stress in schistosomiasis causes DNA damage can be explained by the estrogen-DNA adduct mediated pathway. Estrogen-like metabolites secreted by *S. haematobium* worms and eggs are highly genotoxic as a result of catechol estrogens such as 2-hydroxy(OH)E1(E2) and 4-OHE1(E2) oxidizing to form catechol estrogen quinones and semiquinones such as E1(E2)-3,4Q [123,124]. This results in redox cycling and the production of ROS, which react with DNA. Depurinating adducts form, resulting in the generation of apurinic sites, which have the potential to become mutagenic over several cell generations, eventually becoming cancerous [23,124]. In addition to cancer, mutations generated by apurinic sites are linked to autosomal recessive infertility.

## 7. Genetic Control and Schistosome Infection Intensity

As much as effector cells play a primary role in schistosome infection and severity, multiple factors are also involved in schistosome infection, which are in turn complicated by other elements that center on parasitic and environmental facets that enable transmission. Gene control performs a very distinct and major function by controlling certain host-parasite immune responses. The *SM1* locus on the 5q31-q33 chromosome, discovered from a Brazilian isolate using a model-based analysis method, is one of the major genes shown to influence *S. mansoni* infection and play a role in either susceptibility or resistance to infection. Apart from the localization of *SM1* being linked to eosinophilia and IgE production loci, it also participates directly in regulating genes that encode for anti-inflammatory (IL-4, IL-5, IL-9 IL-13) and pro-inflammatory cytokines (IL-12 and IL-23), as well as CSF-IR, which is an interferon regulating factor-1 (IRF-1) that encodes IFN-inducible genes that include IFN-α and IFN-β [125,126]. Based on the *SM1* model, it was shown that 3% of the population was homozygous and more susceptible to high levels of infection, 29% were heterozygous with moderate levels of infection, while 68% were homozygous and resistant [127]. Further analysis showed that a co-dominant gene called *SM2*, located in the 6q22-q23 region, close to the interferon-gamma receptor 1 (*IFNGR1*) gene, which encodes the IFN-γ α chain receptor and controls disease progression by affecting fibrosis and portal hypertension [125,128]. Furthermore, a susceptibility gene associated with severe hepatic fibrosis was discovered, also in the 6q22-q23 region within four other populations and was shown that these polymorphisms may play a role in affecting nuclear factor binding, gene transcription or transcript stability [128]. Schistosomes possess the ability to evade the immune system and live within the host for several years, causing significant damage to tissue whose repair results in accumulated deposits of extracellular matrix proteins that eventually results in hepatic fibrosis. This manifestation of disease in largely enabled by the connective tissues growth factor (CTGF) gene that encodes a profibrogenic molecule produced by endothelial and hepatic stellate cells, myofibroblasts and hepatocytes [128,129]. This gene has been implicated in progressing severe hepatic fibrosis in Brazilian, Chinese, and Sudanese populations either infected with *S. mansoni* or *S. japonicum*. The gene has also been shown to promote adhesion, migration, and proliferation and is upregulated by M2 macrophages and TGF-β1, which is also produced by M2 macrophages [93]. Other candidate genes less dominant than *SM1* have also been identified; located in the 1p21-q23 region, another within the 6p21-q21 region and lastly in the 7q35-q36 region [127]. These have been shown to possess loci, which play potential roles that are either dependent or independent of *SM1* such as the 1p21-q23 region, which is not, and the locus in the 6p21-q21 region that interacts with *SM1*. Their exact roles and links to *SM1* however, are still yet to be confirmed. Additionally, independent studies in a Senegalese population also identified the *SM1* locus in *S. mansoni* infection, therefore reiterating the role of host genetics in schistosome infection [128].

During a genetic association study on a Malian population, it was shown that certain polymorphisms in the *IL13* promoter were associated with increased risk of *S. haematobium* infection. This was due to the observation that a decrease in IL-13 inhibition related to the T/T genotype, which increases the binding of transcription factors such as nuclear factor-activator of transcription (NF-AT), of which the binding to the IL-13 promoter inhibits IL-13 production. This therefore increases risk of infection as IL-13 may have a role in both larval and egg protective immunity. In addition, it has been shown that *STAT6* (the signal transducer activated by the IL-4/IL-13 pathway) contains the rs324013 polymorphism, which was found in patients exhibiting higher levels of infection. The study concluded that the polymorphisms within the *STAT6* and *IL13* gene perform synergistically to control schistosomiasis infection levels [126,128]. A myriad of other candidate genes whose exact mechanisms still need clarity have been identified in relation to various functions that include infection and intensity (*IFNG, IL10, IL13, IL4, IL5, STAT6, CTLA4, FCN2, COLECC11, ABO*, and *RNASE3*), fibrosis and not infection (*APOE, CCN2, HSPA5, IFNGR1, IL22RA2, MAPKAP1, IL1RL1, TNFA, mTOR, AKT2, TGFB1, TGFBR1, TGFBR2, ACVRL1, SMAD9*, and *SMAD3*), both fibrosis and infection intensity (*RNASE3, IL4, IL10, IFNGI*, and *IL13*), as well as fibrosis within multiple populations (*IFNGR1, IL22RA2, CCN2*, and *MAPKAP1*) with the two former genes also present in the *SM2* quantitative trait locus [130]. The identification of such genes provides opportunities to broaden knowledge in the pathogenesis of schistosomiasis and intensify the discovery of new therapeutics, preventive as well curative treatments [128]. In addition, future endeavors could pursue screening endemic populations for susceptibility, thus resulting in the development of more curative, cheaper and targeted treatment.

## 8. Concluding Remarks

Schistosomiasis is an antique debilitating disease of poverty that is largely driven by the presence of antigen produced by eggs laid by schistosome adults entrapped in host tissues. This results in numerous immunological consequences and this burden placed on the host has severe consequences, including oxidative stress and genetic instability (cancer). *S. haematobium* has already been classified as a definite precursor to bladder cancer. Fortunately or unfortunately, all the immunological reactions observed in schistosomiasis are driven by numerous cytokines that interact in a complex network and various studies have shown that the absence or presence of these cytokines could either progress or down-regulate schistosomiasis. Understanding these mechanisms could therefore be key in finding lasting treatments for this disease through drug development.

## Figures and Tables

**Figure 1 ijms-22-07216-f001:**
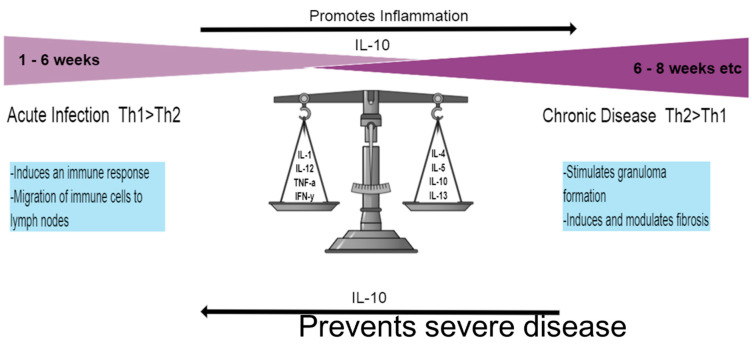
Th1 and Th2 responses. The initial response once infected by the schistosome parasite lasts for ~6 weeks and is dominated by the Th1 response, characterized by pro-inflammatory cytokines. Once the adult worms lay eggs and release SEA, the Th2 response sets in, largely driven by IL-4 and IL-13. Chronic disease requires formation of the granuloma and digresses into liver fibrosis. The Il-10 cytokine represents an important element in disease progression by encouraging inflammation through down-regulation of the Th1 response but also preventing severe disease during the Th2 response.

**Figure 2 ijms-22-07216-f002:**
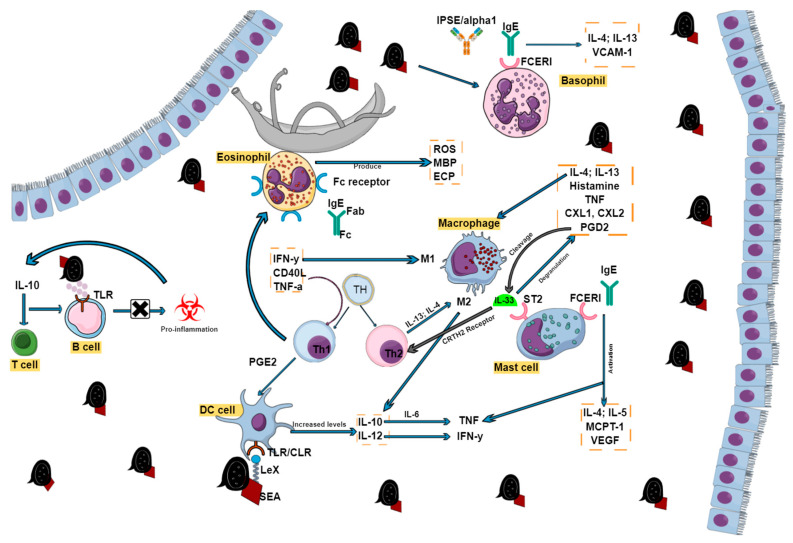
Effector cell activation during schistosome infection. The presence of inflammatory cells in the form of lymphocytes (B and T cells), macrophages, eosinophils, dendritic cells, mast cells, and basophils all play crucial roles in schistosome disease progression. Various receptors and cytokines drive their functions, which may sometimes overlap but are all the same interconnected.

**Figure 3 ijms-22-07216-f003:**
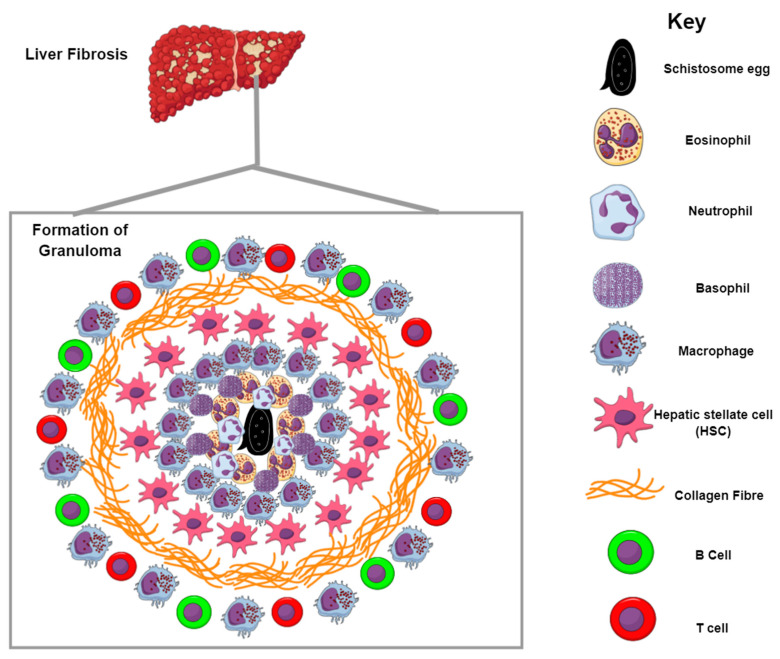
Formation of the granuloma and liver fibrosis. Granulomas are formed by the recruitment of inflammatory cells to the site of infection. Schistosome eggs release SEA that trigger the generation of antibodies, which recruit inflammatory cells mostly in the form of macrophages and eosinophils around the eggs. Neutrophils and basophils can also be seen during the initial phase of granuloma formation. The contents of the granuloma are bound by collagen fibers whose production is largely driven by IL-13; this subsequently results in liver fibrosis.

**Figure 4 ijms-22-07216-f004:**
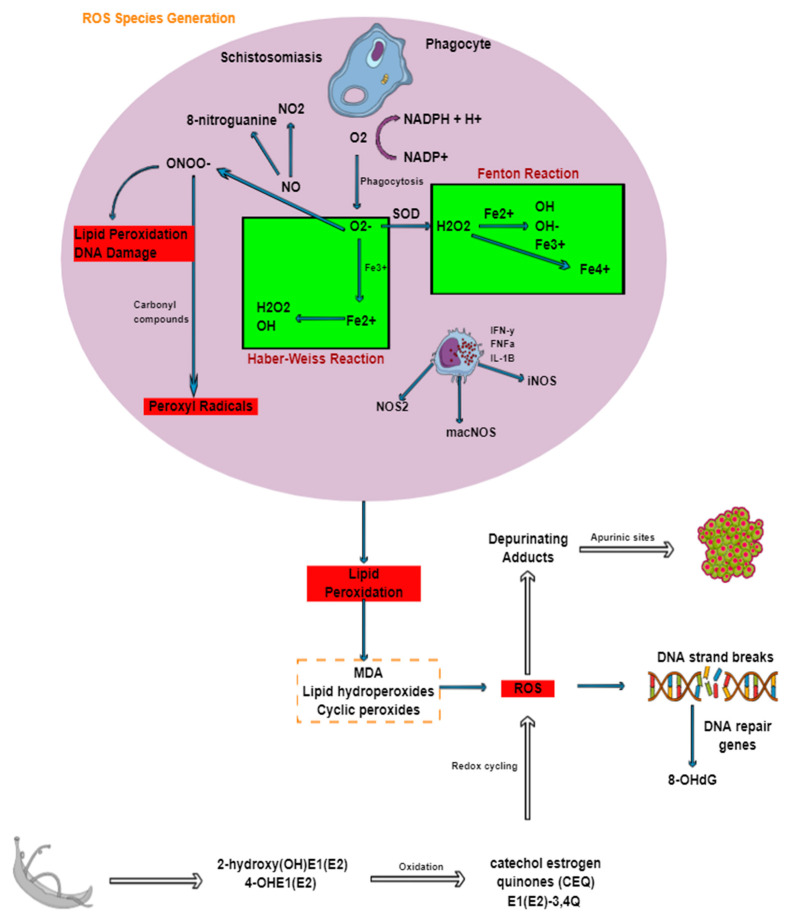
Generation of ROS during schistosomiasis and its association with DNA damage and cancer. Through oxidative or respiratory burst ROS are formed. Nitrosative stress is primarily produced from macrophages and nitric oxide, which reacts with superoxide anion. The production of ROS results in lipid peroxidation, which eventually leads to breaks in DNA strands and subsequently cancer.

## Data Availability

Not Applicable.

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
