# Peer review of "Immunological and Biochemical Interplay between Cytokines, Oxidative Stress and Schistosomiasis"

_ijms, 2021, doi:10.3390/ijms22137216_

Round 1
Reviewer 1 Report
The review paper by Masamba and Kappo entitled: “Immunological and biochemical interplay between cytokines, oxidative stress and Schistosomiasis” is anice paperthat offers a view of the molecular interactions between the parasite and the host. Although numerous revisions on Schistosoma are available in the literature in the last decade, the present one goes in depth in the cytokine network.
Although the paper is well written, in some parts it is difficult to follow for non-expert readers, since it has plenty of scientific terms less frequently used by non-expert (immunologist, chemistry experts...) . I would recomment the authors to simplify some sentences for a better understanding (and a more fluid reading).
In my opinion, the present paper deserves publication, but several minor issues should be revised by the authors.
Lines 57-58: Please rephrase this sentence to ease the reading: “These actions include the down-regulation of pro-inflammatory and up-regulation of anti-inflammatory cytokines”
Lines 278-306. Again, to ease the reading, I recommend authors to employ the same terminology for activated macrophages and call them M1 and M2 instead of CAMɸS or AA macrophages/AA phenotype along the paragraph, after clarifying the nomenclature (line 289-304, and later on in the manuscript). In Figure 2, these cells also appear as M1 and M2, and is some other parts of the article. Please, be consistent with the nomenclature.
Figure 2 Footer: the text mention the presence of B lymphocytes but they are not present in the scheme. There are antibodies, but not B cells represented. Maybe the authors should substitute the term B and T cells for the word lymphocytes. Also, within the figure, there is a mistake and the box at the right side of macrophages PG2 instead of PGD2 is written.
Lines 454-456: the recruit of neutrophils is not represented in fig. 3, but it is mentioned in the text. A small amount of basophils can be found at the initial phase of the granuloma formation as well. Figure 3 represents and advanced granuloma, in which the abovementioned cells are not present (while they are in the context of initial granuloma development). This fact should be mentioned in the figure or, alternatively, the stage of the granuloma in figure 3 should be identified.
The objective of the present paper is to illustrate the relations between cytokines in Schistosome infections. Many of the reactions are attributed to different antigens (SEA) but other molecules could have an important role and they should be briefly mentioned. For instance, I missed some components of exosomes, like miRNA (Liu et al 2019 and others).
Minor. Line 119, please, substitute “specie” for “species”
Reviewer 2 Report
Dear authors,
The review is complete, but the manuscript has many writing errors, grammatical errors present not only in the background of the text but also in the form. A couple of examples:
Lane 165: replace “naïve” with “naive”
Lane 167, replace “dentritic” with “dendritic”.
Names like mycobacteria should not be italicized, only genus like Mycobacterium
I suggest that the entire manuscript be sent to correction of style because it is challenging to read. It takes a lot of work to review each paragraph.
About the figures:
Although the drawings are illustrative, criteria should be "standardized": sizes, colors, letters within the tables. Some tables have such intense colors that they make reading difficult. The dimensions of the words in the graphs and the typefaces must also be standardized.
